# Molecular Cloning and Expression Analysis of the Typical Class III Chitinase Genes from Three Mangrove Species under Heavy Metal Stress

**DOI:** 10.3390/plants12081681

**Published:** 2023-04-17

**Authors:** Yue-Yue Zhou, You-Shao Wang, Cui-Ci Sun

**Affiliations:** 1State Key Laboratory of Tropical Oceanography, South China Sea Institute of Oceanology, Chinese Academy of Sciences, Guangzhou 510301, China; 2Daya Bay Marine Biology Research Station, Chinese Academy of Sciences, Shenzhen 518121, China; 3Innovation Academy of South China Sea Ecology and Environmental Engineering, Chinese Academy of Sciences, Guangzhou 510301, China; 4University of Chinese Academy of Sciences, Beijing 100049, China

**Keywords:** chitinase, mangrove plants, cloning, bioinformatic analysis, gene expression, heavy metal

## Abstract

Chitinases are considered to act as defense proteins when plants are exposed to heavy metal stresses. Typical class III chitinase genes were cloned from *Kandelia obovate*, *Bruguiera gymnorrhiza*, and *Rhizophora stylosa* by using RT-PCR and RACE and named *KoCHI III*, *BgCHI III,* and *RsCHI III*. Bioinformatics analysis revealed that the three genes encoding proteins were all typical class III chitinases with the characteristic catalytic structure belonging to the family GH18 and located outside the cell. In addition, there are heavy metal binding sites in the three-dimensional spatial structure of the type III chitinase gene. Phylogenetic tree analysis indicated that CHI had the closest relationship with chitinase in *Rhizophora apiculata.* In mangrove plants, the balance of the oxidative system in the body is disrupted under heavy metal stress, resulting in increased H_2_O_2_ content. Real-time PCR illustrated that the expression level under heavy metal stress was significantly higher than that in the control group. Expression levels of *CHI III* were higher in *K. obovate* than in *B. gymnorrhiza* and *R. stylosa*. With the increase in heavy metal stress time, the expression level increased continuously. These results suggest that chitinase plays an important role in improving the heavy metal tolerance of mangrove plants.

## 1. Introduction

Mangroves are a wetland woody plant community that grows in the intertidal zone of tropical and subtropical coasts and also play an important role in purifying seawater, preventing wind and waves, storing blue carbon, and maintaining biodiversity [1,2,3]. In recent years, with the rapid development of industry and agriculture and the expansion of coastal cities, large amounts of sewage and waste are discharged into estuaries and coastal areas every year, and the continuous development of the marine transportation industry causes water pollution in the bay area. Environmental pollution in the mangrove growing area has become an environmental problem that cannot be ignored, especially heavy metal pollution as a typical persistent pollutant; heavy metals can not only endanger the security of the mangrove ecosystem, but also pass through the food chain to other consumers and continuously accumulate in organisms, eventually posing a threat to the health of human beings and other organisms. This harm should not be ignored. Heavy metal pollution in estuaries and bays has become increasingly acute, which has attracted the attention of ecologists and environmental protection departments [4,5]. Excessive metal can inhibit seed germination and seedling growth, destroy antioxidant enzymes and membrane systems, cause chromosomal aberrations, and lead to plant death. Moreover, metal ions also seriously interfere with plant metabolism by causing secondary stresses such as nutritional imbalance, generating large amounts of free radicals (ROS) and oxidative stress [6,7]. At present, it is believed that the adaptation mechanism of plants to heavy metal roughly includes: (1) the root effect and efflux of heavy metals; (2) the chelation of compounds; (3) the regulation of the antioxidant system; (4) and the induction of related gene expression [8]. Although many available data show that some mangrove plants appear to possess a great tolerance for high levels of heavy metal pollution [9,10], very little information exists on the molecular mechanisms of their heavy metal tolerance.

The chitinase gene is a typical plant defense gene that plays an important role in plant growth and development, stress resistance, and defense response [11,12,13,14]. Chitinase is divided into five types I–V and four families (18, 19, 23, and 48 families) [15,16,17]. Chitinases are responsible for the hydrolysis of beta-1,4-linkages in chitin [18]. The GH18 family is widely distributed and found in viruses, bacteria, plants, fungi, and animals [19]. Chitinases in plants are encoded by single genes. Some types of chitinase are localized in extracellular spaces and some types of chitinase are localized in intracellular spaces. Most of them are small molecular proteins with a molecular weight between 25 and 35 kDa [20], and are usually very acidic or very alkaline. The type III chitinases in plants are similar to those of fungi and bacteria, and do not have the chitinase binding domain (CBD) [21,22,23]. When plants are induced by inducible factors (polysaccharides such as chitin, fungal cell walls as chitin) and some stresses such as ethylene and heavy metals, chitinase activity increases, which enhances plant defenses [24,25,26]. Plants have evolved multiple defense mechanisms to respond to metal toxicity, and chitinases are involved in the plant defense response to metal stress. The chitinase genes in *Viciafaba*, *barley*, *maize*, and *soybean*, induced by lead, arsenic, and cadmium, showed that this enzyme has a role in preventing heavy metal toxicity [24,27]. It has been shown that multiple isoforms of chitinase were detected in the roots of two soybean (*Glycine max*) varieties, *Chernyatka* (metal-ion-resistant) and *Kyivska 98* (metal-ion-sensitive). The expression patterns of the two soybean root isozymes differed after different metal ion stresses; for example, isoform D expression was up-regulated in *Chernyatka* after Cd^2+^ stress, while isoform D expression was not significantly changed in *Kyivska 98* [28]. In addition, metal ions such as Na^+^ and Fe^3+^ have been reported to increase chitinase activity, while Zn^2+^ and Co^2+^ showed inhibitory effects on chitinase activity [29]. At present, only two full-length cDNA sequences of chitinase genes (class I and III) have been reported from mangrove plants of *Avicennia marina* and *Aegiceras corniculatum* in our previous research [30,31].

In the paper, three mangrove species, *Kandelia obovate*, *Bruguiera gymnorrhiza*, and *Rhizophora stylosa,* were selected for the experiments. PCR techniques, genomic DNA library construction, gene cloning techniques, and the RACE method were used to carry out full-length cloning of type III chitinase genes of mangrove plants, and bioinformatics was applied to analyze the primary structure, secondary structure, tertiary structure, and gene function analysis of the genes. In this study, mangrove plants were subjected to stress treatment with complex heavy metals (Cu, Cd, and Pb) commonly found in the environment to study in detail the expression response of a particular chitinase gene. Additionally, the expression characteristics of the genes under the stress of heavy metals were analyzed via real-time fluorescence quantitative polymerase chain reaction technology. This is the first time the full-length cDNA of class III chitinase genes (*BgCHI III*, *KoCHI III*, and *RsCHI III*) has been cloned from *B. gymnorrhiza*, *K. obovate*, and *R. stylosa* to explore the potential molecular mechanism of mangrove plant tolerance to heavy metal pollution. This study will provide more details on the molecular mechanisms of or a scientific basis for coastal wetland heavy metal environmental remediation using mangrove plants.

## 2. Results

### 2.1. The Full-Length cDNA of the CHI III Gene Cloning

In Figure 1A, the integrity of the extracted total RNA is good. The OD_260_/OD_280_ values of the total RNA samples were between 1.8 and 2.1, indicating that the extracted total RNA had a high purity. After obtaining the intermediate fragment, the nucleotide sequences were obtained via sequencing. The nucleotide sequences were carried out in the NCBI database via BLAST comparison, and the similarity with the existing chitinase sequence reached 99%, which indicated that it was a chitinase sequence. The 3 and 5′ end-specific fragments (Figure 1B) were obtained by using the RACE method. After sequencing, the full length was finally determined after splicing.

Through SMART^TM^ RACE cDNA amplification, the full-length cDNA sequence of chitinase was isolated from mangrove and named *BgCHI III* (Figure 2A), *KoCHI III* (Figure 2B), and *RsCHI III* (Figure 2C). The nucleotide sequence of *BgCHI III* was 1067 bp, containing a 711 bp open reading frame (ORF) encoding the deduced protein length of 236 amino acids. The ORF starts at the 109 ATG start codon and ends at the 819 TGA stop codon (Figure 2A). The calculated molecular weight of the protein was 25.3 kDa and the isoelectric point was 6.06. Amino acid composition analysis showed that *BgCHI III* contained higher levels of Ser (11.4%), Gly (11.0%), and Ala (8.9%). The stability coefficient was 43.50, and the hydrophilic coefficient was −0.252.

The nucleotide sequence of *KoCHI III* was 1071 bp, containing a 600 bp open reading frame (ORF) encoding the deduced protein length of 199 amino acids. The ORF starts at the 225 ATG start codon and ends at the 824 TGA stop codon (Figure 2B). The calculated molecular weight of the protein was 21.7 kDa and the isoelectric point was 5.69. Amino acid composition analysis showed that *KoCHI III* contained higher levels of Ser (11.6%), Gly (10.6%), and Ala (8.5%). The stability coefficient was 42.31, and the hydrophilic coefficient was −0.284.

The nucleotide sequence of *RsCHI III* was 1063 bp, containing a 600 bp open reading frame (ORF) encoding the deduced protein length of 199 amino acids. The ORF starts at the 216 ATG start codon and ends at the 815 TGA stop codon (Figure 2C). The calculated molecular weight of the protein was 21.6 kDa and the isoelectric point was 5.69. Amino acid composition analysis showed that *RsCHI III* contained higher levels of Ser (11.6%), Gly (10.6%), and Ala (8.5%). The stability coefficient was 41.98, and the hydrophilic coefficient was −0.285.

Analysis indicated that the protein was an unstable hydrophilic protein with no transmembrane helix signals. Subcellular localization predictions using the online software Plant-mPLoc [32] pairs showed that three genes encoding type III chitinase proteins may be non-secretory proteins localized extracellularly, which may be related to the defense mechanism of chitinase in mangrove plants. This is consistent with the previous conclusion that type I and V chitinases are located in the vacuole, and type III chitinase is located extracellularly. In addition, chitinase belongs to the 18th family of glycohydrolase (GH18) in terms of structure. Without a chitinase binding domain (CBD), it is a typical type III chitinase. Secondary structure analysis showed that the chitinase consisted of seven α-helices and seven β-turns (Figure 3). Based on SWISS-MODEL analysis [33], ribbon cartoon and space-filling models of *CHI III* are presented in Figure 4. Taking the three-dimensional spatial structure of *pomegranate* as a template, the three-dimensional spatial structure of the mangrove was established. GH 18 chitinase from *pomegranate* has the crystal structure of *pomegranate* chitinase III. In this study, there are heavy metal binding sites in the three-dimensional spatial structure of the type III chitinase gene. The type III chitinase gene may combine with heavy metal ions to reduce the entry of heavy metal into cells, reducing the toxicity of heavy metals to plant cells. This also provides insights on its metal storage capacity.

The phylogenetic tree was constructed via the NJ method using MEGA6 software. In this study, we chose *Vitis vinifera* (AB105374), *Arabidopsis thaliana* (NM-122314.3), *Glycine max* (NM-001249780.1), *Rhizophora apiculate* (KY795335.1), *Hevea brasiliensis* (DQ873889.2), *Citrullus colocynthis* (JQ756124.1), *Citrullus lanatus* (DQ180495.1), *Nepenthes rafflesiana* (GQ338259.1), *Avicennia marina* (JQ655770), *Aegiceras corniculatum* (JQ655771), *Bruguiera gymnorrhiza*, *Kandelia obovate*, and *Rhizophora stylosa*. The amino acid sequences of chitinase type III in these plants were selected to construct the phylogenetic tree. The phylogenetic analysis results show that the chitinase type III genes of *B. gymnorrhiza*, *K. obovate*, and *R. stylosa* belonging to *Rhizoporaceae* were clustered together, while the chitinase type III genes of *A. marina* belonging to Myrsinaceae and *A. corniculatum* belonging to Verbenaceae evolved another branch that was quite different. Phylogenetic tree analysis indicated that CHI had the closest relationship with chitinase in *Rhizophora apiculate* (94.40% similarity) (Figure 5). Although *B. gymnorrhiza*, *K. obovate*, and *R. stylosa* belong to oriental mangrove species and *R. apiculate* belongs to western mangrove species, their chitinase genes are closely related.

### 2.2. The Effect on Mangrove under Heavy Metal Treatment

A plant’s growth is slowed by heavy metal treatment, resulting in yellowing of the leaves. The heavy metal exposure affects the morphology of the mangrove plants. In this study, we observed that the leaf morphology of *K. obovate* and *R. stylosa* changed after exposure to high concentrations of heavy metals (Figure 6). The content of H_2_O_2_ can reflect the accumulation of ROS in plants [34]. The changing trend in H_2_O_2_ content was obvious with the increase in the concentration of composite heavy metals. At the highest concentration of treatment, the H_2_O_2_ content of the five mangrove plants increased significantly compared with the control group. H_2_O_2_ content was 3.78 times higher than the control of *B. gymnorrhiza* at C3. H_2_O_2_ content was 6.17 times higher than the control of *K. obovate* at C4. H_2_O_2_ content was 7.57 times higher than the control of *R.stylosa* at C4. The order of H_2_O_2_ content was *K. obovate* > *R. stylosa* > *B. gymnorrhiza* (Figure 7).

### 2.3. CHI III mRNA Expression in Leaf in Response to Heavy Metals

The effects of heavy metals on the expression of *CHI III* mRNA in leaves are shown in Figure 8. Moreover, there were different patterns of *CHI III* mRNA expression response to heavy metal stress among the three mangrove plants. After 3 days of exposure, *BgCHI III* mRNA expression was 2.41 times higher than the control at C2. *BgCHI III* mRNA expression was 1.97 times higher than the control at C4. In addition, the maximum expression level was detected at C2 after 3 days in *B. gymnorrhiza* leaves (Figure 8A). After 28 days of exposure, *KoCHI III* mRNA expression was 14.79 times higher than the control, and the maximum expression level was detected at C2 (Figure 8B). Under the treatment of heavy metal C4 concentration, the expression levels of *R. stylosa* were relatively high, reaching a maximum value of 6.74 times that of the control group on the 28th day of treatment (Figure 8C). Under treatment at C2 and C4 concentrations and with the increase in heavy metal stress time, the expression change had an upward trend in *K. obovate*. Under C1, C3, and C4 heavy metal treatment, the expression change had an upward trend in *R. stylosa* with the increase in heavy metal stress time. The expression levels of the CHI gene with C2 stress treatment in K. obovate on the 28th day of treatment were higher than those in *R. stylosa* and *B. gymnorrhiza*.

## 3. Discussion

### 3.1. Cloning and Structural Characterization Analysis of CHI III

In this study, chitinase genes were cloned for the first time from *B. gymnorrhiza*, *K. obovate*, and *R. stylosa*. These were predicted and analyzed for their secondary structure, subcellular localization, transmembrane domain, and functional domain of the cloned amino acid sequences. The amino acid sequences of chitinases from other plants also underwent multiple comparisons and were phylogenetically analyzed to demonstrate the homology between the cloned genes and chitinase genes from other plants. In the process of molecular evolution, amino acids with strong functions in the protein components evolved slowly, so the base evolution of amino acids with strong functions maintained relatively high conservation. Generally, the homology of sequences is closely related to the similarity of their functions, so the homology sequence of chitinase cloned via NCBI can be used to predict the function of the target gene according to its homology sequence. Constructing a phylogenetic tree can help us analyze protein evolution more clearly. The chitinases of glycoside hydrolase family 18 and glycoside hydrolase family 19 differ greatly in structure and distribution due to their different origin and evolution [34]. GH18 family chitinases include class III and V chitinases in plants [35,36]; class III and V chitinases have both lysozyme activity and chitinase activity [24]. In terms of structure, the 18 family only contains one catalytic region. In terms of distribution range, the glycosidase 18 family is widely distributed in microorganisms, animals, and plants. In our study, the cloned chitinase genes of mangrove plants were searched using Blast in NCBI, and it was found that the gene was highly similar to the chitinase gene of other plants, indicating that the three genes cloned were chitinase genes. Bioinformatics analysis showed that the chitinase gene encodes a typical type III chitinase that belongs to the glycoside hydrolase family 18. A conserved catalytic region means that it has a glycohydrolase function. Without CBD structure, it does not have anti-bacterial function in vitro. The chitinase protein is located outside the cell. When it encounters pathogens, it can directly hydrolyze new hyphae and resist pathogenic fungi. The sequences of type III chitinase genes from 11 other plants were selected for comparison in this chapter, and the three cloned type III chitinase genes were found to be the most evolutionarily similar (94.40% similarity) to the type III chitinase gene of *R. apiculata*. All three mangrove species belong to the family *Mangroveaceae*; the developmental tree branches were clustered together, and the kinship determination was based on amino acids. The results of kinship determination based on sequence analysis were the same as those based on traditional evolutionary kinship determination. The first basis for the investigation of gene function is to know the full-length sequence of the gene and the structural features of the protein space.

### 3.2. Expression of CHI III in Leaves in Response to Heavy Metal

Plant chitinases are disease-course-associated proteins that can disrupt the cell walls of pathogenic bacteria and the exoskeletons of insects and are also regulated by other disease-resistance-related molecules such as salicylic acid (SA), which play an important role in plant defense processes and are thus widely used in the genetic engineering of plant disease resistance [37,38,39,40]. In addition, chitinases are involved in plant responses to abiotic stresses such as drought, metal ions, and high temperatures. When plants were subjected to environmental stress, type III chitinase showed strong induction and expression [28,40]. The cotton type III chitinase gene is *GhaChi,* which is induced by ABA [41]. The sugarcane type III chitinase gene plays a positive role in response to biotic and abiotic stresses [26]. When *Arabidopsis* plants were subjected to environmental stress, especially salt stress and mechanical damage, only class III chitinase genes were induced and expressed [37]. Zhou’s study shows that *SlChi* is a gene responsive to salt stress in *Salix*. When *Salix* was treated with 100 mmol/L NaCl stress, the expression level of endogenous *SlChi* increased with time, and the gene expression level after 48 h of continuous high salt treatment was 40 times that of non-treatment [42]. Furthermore, it has now been shown that overexpression of chitinase in transgenic plants does improve resistance to heavy metals [43]. Previous studies have focused on a single phenotype of the gene and studied the differences in its expression in different tissues, lacking the analysis of multiple phenotypes of the gene in multiple species. Among them, the mechanism of action of chitinase in mangrove plants under heavy metal stress is less studied, and one study cloned and compared the differences between different tissues of the type I chitinase of *A. corniculatum* [31]. In this study, when *K. obovate* was treated with C2 heavy metal stress, the expression level of endogenous *KoCHI* increased with treatment time, and the expression level of the gene after 28 days of heavy metal treatment was 14.49 times that of non-treatment. Additionally, the expression level of the gene after 28 days of heavy metal treatment was 6.47 times higher in *R. stylosa* treated with C4 heavy metal stress. The *CHI III* expression in leaves was up-regulated with heavy metals, which was in good agreement with previous results in other plants [20,44]. In nature, mangrove plants are stressed by various heavy metals. When the concentration of heavy metal is too high, it causes a series of injuries and cells produce a series of stress responses [45,46]. At present, it is generally believed that plants have an innate “immune system”, oxidative burst (OXB), meaning that plant cells rapidly produce reactive oxygen species (ROS) under stress conditions that initiate other signal cascades in the body [47,48]. In most cases, H_2_O_2_ is the main accumulation product of OXB [49,50]. In the present study, the H_2_O_2_ content accumulated more after heavy metal stress treatment, and according to previous studies, many heavy metals entering plant cells would cause the accumulation of ROS, and the antioxidant system in plants would act as a regulator to convert ROS into H_2_O_2_, which would be a signaling factor to trigger the expression of some genes when accumulated in excess [51,52]. Our experiments to understand the structural features of the chitinase gene properties and protein space revealed that type III chitinase acts mainly outside the cell and has metal binding sites that may bind directly to heavy metals. The possible intrinsic molecular mechanism is that heavy metal stress leads to high levels of H_2_O_2_ in vivo, and H_2_O_2_ becomes a signaling factor triggering the expression of chitinase, which may bind to heavy metals and attenuate heavy metal damage to plant leaves (Figure 9). There are different phenotypes of chitinases, and the functional roles they have can vary, as other reports say that chitinases are considered components of the second line of defense under metal stress [53]. They may alter the kinetics and permeability of the cell wall to metals, as well as affect the metal binding and fixation capacity of the cell wall [54]. They seem to be stable components of the plant metal stress defense [27]. Indeed, transgenic plants overexpressing chitinase have shown increased tolerance to metal [43].

Overall, expression of the chitinase gene may have improved the tolerance and accumulation of heavy metals in mangrove plants. In coastal wetlands, mangrove plants can be used as remediation plants for heavy metal pollution. Therefore, it is speculated that CHI may play a role in the resistance of mangrove plants to heavy metal stress.

## 4. Materials and Methods

### 4.1. Plant Materials and Treatments

The hypocotyls of mature *B. gymnorrhiza*, *K. obovate,* and *R. stylosa* were collected from the Leizhou Peninsula, Guangdong Province, China, with salinity ranging from 22‰ to 30‰. Each pot was irrigated with 500 mL of 1/2 Hoagland’s nutrient solution (containing 10‰ NaCl) every 3 days.

When the plants grew to 3–5 true leaves, the mangrove seedlings were irrigated with 1/2 Hoagland’s nutrient solution containing mixed heavy metal sewage of different pollution levels (Table 1). C1: 5 mg/L CuCl_2_, 1 mg/L PbCl_2_, 0.2 mg/L CdCl_2_; C2: 25 mg/L CuCl_2_, 5 mg/L PbCl_2_, 1 mg/L CdCl_2_; C3: 50 mg/L CuCl_2_, 10 mg/L PbCl_2_, 2 mg/L CdCl_2_; C4: 75 mg/L CuCl_2_, 15 mg/L PbCl_2_, 3 mg/L CdCl_2_. Watered once every 3–5 days, each treatment was 600 mL, and then combined heavy metal treatment sampling was performed simultaneously at 0 days, 3 days, 7 days, 14 days, and 28 days. The leaves of control and treatment plants were quickly placed in liquid nitrogen and stored at −80 °C on standby. After 28 days of treatment, plant photographs of morphological changes were recorded and H_2_O_2_ content was determined. The amount of hydrogen peroxide can be detected by measuring the reaction product of H_2_O_2_ and molybdic acid [55].

### 4.2. Extraction of the Total RNA

The total RNA of mangrove leaves was extracted using the TIANGEN polysaccharide polyphenol plant total RNA extraction kit centrifugal column method. The extracted RNA was analyzed for purity and integrity. The RNA was treated with DNase I, and Oligo(dt)_15_ was used as the universal primer. The treated RNA was reverse transcribed with reverse transcriptase to synthesize first-strand cDNA for RT-PCR amplification.

### 4.3. Primer Designs and Sequence Comparison Analysis

The amino acid sequences of chitinase genes from five closely related plants were downloaded from NCBI, and the conserved sequences were found using BioEdit sequence software. The primers F1 and R1 were designed with Primer Premier 5.0 using the closest sequence as a template.

### 4.4. Cloning the Full-Length cDNA of the Chitinase Gene

The intermediate fragment was cloned via PCR with the first-strand cDNA as the template and the primers F1, R1 (Table 2). The intermediate fragment sequence of the chitinase gene was obtained. Primers GSP-3′, GSP-5′, NGSP-3′, and NGSP-5′ were designed according to the obtained intermediate fragment sequence (Table 2). The specific primers were designed using the software Primer Premier 5.0. The full-length cloning was carried out via the RACE method according to the instructions of the “SMARTTM RACE cDNA Amplification Kit” from CLONTECH. During RACE, the reverse transcription product was used as a template. The outer primer was used for amplification, and the inner primer was used for nested PCR amplification. Additionally, single primer control was set. The PCR products were recovered using a GenStar kit and ligated with pMD18-T vector. The ligation product was transformed into *E. coli* DH5α competent cells, and two-way sequencing was completed by the HuaDa Gene Sequencing Company. We obtained the full-length cDNA using Contig Express software. The sequencing results were searched and analyzed in GenBank.

### 4.5. Bioinformatic Analysis

The full-length sequence was subjected to bioinformatics analysis using the following software or online tools: validation of the full-length sequence and analysis of the functional domains (Table 3).

### 4.6. Analysis of Gene Expression using Real-Time Quantitative PCR

The preparation method of first-strand cDNA is shown above. The specific primers for the type III chitinase gene used in semi-quantitative RT-PCR were F and R (Table 2). The primers Bg18S (F), Bg18S (R), Ko18S (F), Ko18S (R), Rs18S (F), and Rs18S (R) (Table 2) were designed according to the RNA18S histone of mangrove as the cDNA content reference and for DNA contamination detection. Methods: NTC control was carried out on each plate of each gene, and qPCR96Well was taken to prepare the following reaction system (Table 4). Fold induction in CHI III mRNA expression relative to the control was determined by the standard 2 -DDCT method of Livak and Schmittgen [56].

### 4.7. Statistics Analysis

Using statistical analysis software, significant differences between controls and treatments were determined via one-way ANOVA and analyzed using SPSS V.22.0 software (Dunnett’s multiple comparison test). All experiment data are presented as mean ± standard values of three independent sets.

## 5. Conclusions

Three chitinase genes (CHI) were cloned for the first time from *B. gymnorrhiza*, *K. obovate,* and *R. stylosa*; these belong to the 18th glycosidase family and are class III chitinases. Our results indicate that leaf *CHI III* expression was regulated by Cu, Cd, and Pb. After exposure to heavy metal, *CHI III* mRNA was strongly activated. Moreover, *CHI III* consistently presented the highest transcript level after 28 days of exposure. There was a higher expression level in *K. obovate* compared with others. This study will open a window on the function of the chitinase gene and provide a solid basis for the mechanism of heavy metal resistance in mangroves.

## Figures and Tables

**Figure 1 plants-12-01681-f001:**
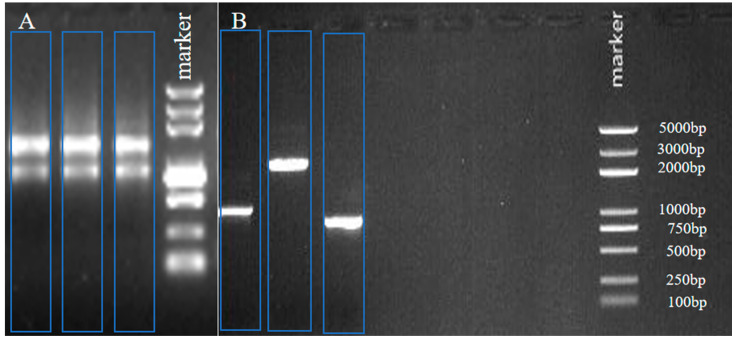
Agarose gel electrophoresis of total RNA (**A**), PCR products of 3′ or 5′ RACE (**B**).

**Figure 2 plants-12-01681-f002:**
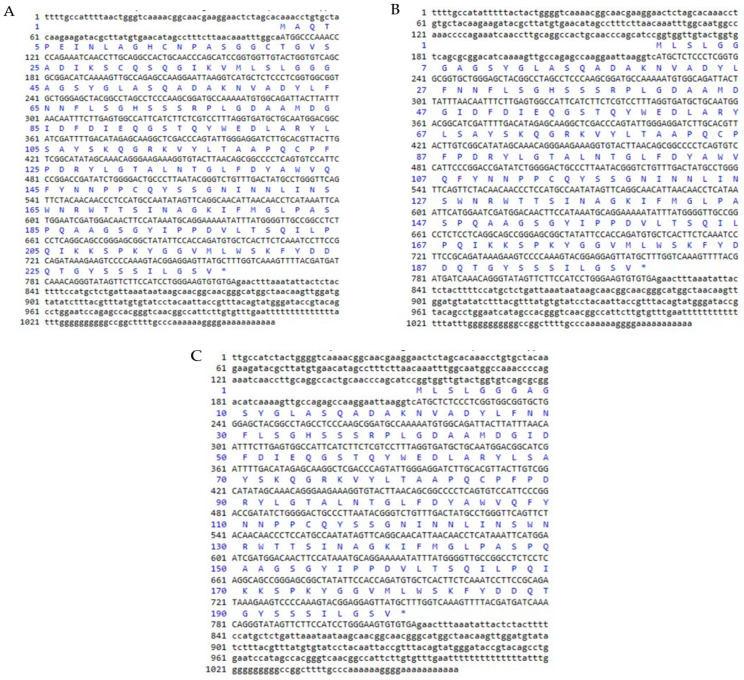
CHI gene cDNA sequence and its encoded amino acid sequence. The encoded protein starts from start codon ATG to stop codon TAG, and the predicted amino acid sequence is shown in one-letter code under the DNA sequence. Complete gene bands contain AATAA-box and poly (**A**) tail. (**A**) *BgCHI*, (**B**) *KoCHI*, (**C**) *RsCHI*. “*” represents the position of the end of protein translation in the figure.

**Figure 3 plants-12-01681-f003:**
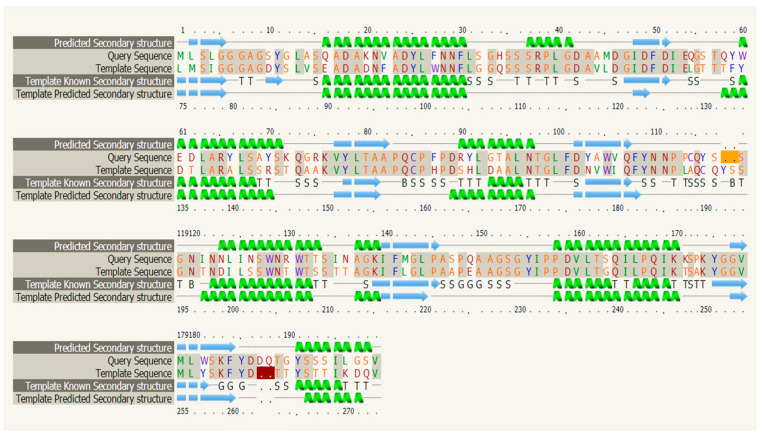
Prediction of secondary protein structure.

**Figure 4 plants-12-01681-f004:**
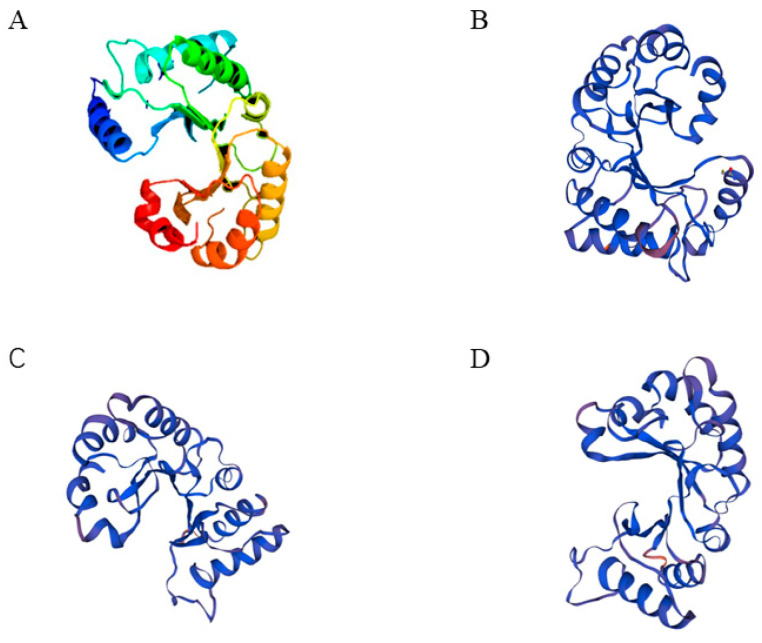
The molecular model of CHI III. (**A**) Pomegranate. (**B**) Bruguiera gymnorrhiza. (**C**) Kandelia obovate. (**D**) Rhizophora stylosa.

**Figure 5 plants-12-01681-f005:**
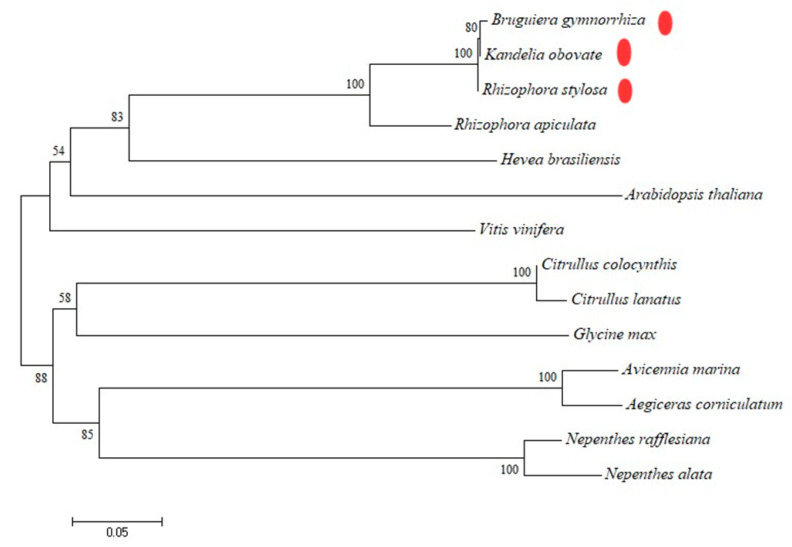
Phylogenetic tree of the CHI III. Multiple alignments of the sequences of CHI III and other selected plant chitinase were performed using MEGA 6. The three red point represents the chitinase of three mangrove plants in the figure.

**Figure 6 plants-12-01681-f006:**
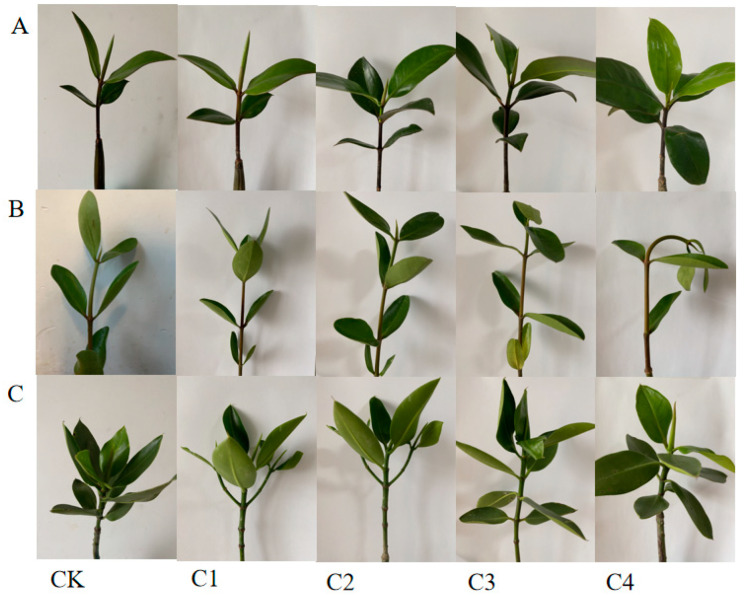
Morphological changes under heavy metal treatment of mangrove plants. (**A**) *Bruguiera gymnorrhiza.* (**B**) *Kandelia obovate.* (**C**) *Rhizophora stylosa*.

**Figure 7 plants-12-01681-f007:**
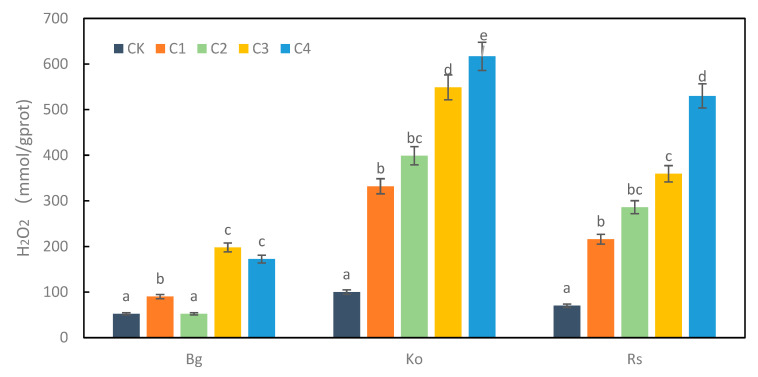
Effect of heavy metal treatments on H_2_O_2_ content of mangrove plants. Data are the means ± standard of three separate individuals. Each bar with different lowercase letters was significantly different (*p* < 0.05).

**Figure 8 plants-12-01681-f008:**
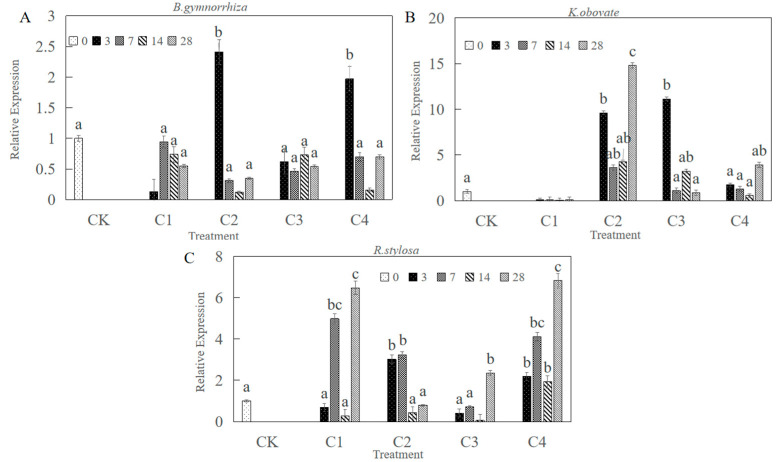
Expression of CHI gene in leaves of three species in response to heavy metal stresses using real-time quantitative PCR analysis. (**A**) *Bruguiera gymnorrhiza.* (**B**) *Kandelia obovate.* (**C**) *Rhizophora stylosa*. Data are the means ± standard of three separate individuals. Each bar with different lowercase letters was significantly different (*p* < 0.05).

**Figure 9 plants-12-01681-f009:**
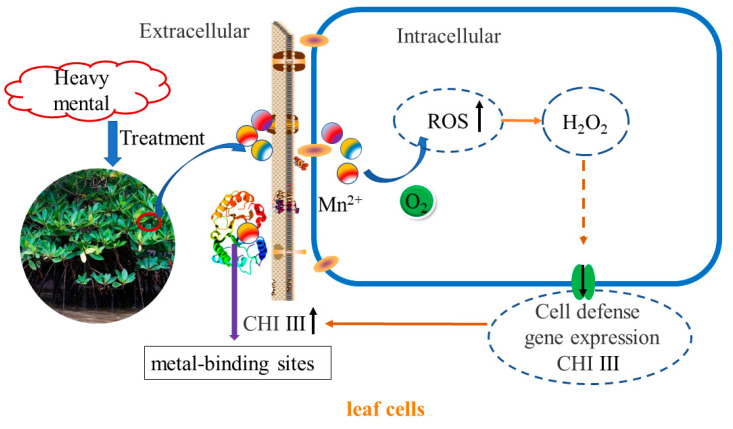
Concept map of chitinase action mechanism of mangrove plants under heavy metal stress.

**Table 1 plants-12-01681-t001:** Heavy metal concentrations in artificial sewage prepared from 1/2 Hoagland’s nutrient solution.

Heavy Metal (mg/L)	Control Group (CK)	C1	C2	C3	C4
Cu^2+^	0	5	25	50	75
Pb^2+^	0	1	5	10	15
Cd^2+^	0	0.2	1	2	3

**Table 2 plants-12-01681-t002:** List of primers for PCR, RACE, and real-time PCR experiments.

Primers	Sequence (5′–3′)
F1	TTGCCATCTACTGGGGTCA
R1	CGTACTTTGGGGACTTCTTTAT
UPS	CTAATACGACTCACTATAGGGC
GSP-3′	GATTACGCCAAGCTTGGGTCAAAACGGCAACGA
GSP-5′	GATTACGCCAAGCTTGTAAAACGACGGCCAGTG
NGSP-3′	GATTACGCCAAGCTTTACGGGTCTGTTTGACTATG
NGSP-5′	GATTACGCCAAGCTTATCTGCGGAAGGATTTGA
F	GGACATCAAAAGTTGCCAGAG
R	GCATCACCTAAAGGACGAGAA
Bg18S(F)	CGGGGGCATTCGTATTTC
Bg18S(R)	CCTGGTCGGCATCGTTTAT
Ko18S(F)	CCTGAGAAACGGCTACCACATC
Ko18S(R)	ACCCATCCCAAGGTCCAACTAC
Rs18S(F)	ACCATAAACGATGCCGACC
Rs18S(R)	CCTTGCGACCATACTCCC

**Table 3 plants-12-01681-t003:** Methods and the website of bioinformatics analysis.

Function	Tool
Spliced sequence alignment	ApE software
Predict the open reading frame	ORFFinder (https://www.ncbi.nlm.nih.gov/orffinder/ accessed on 15 October 2021)
Predict the molecular weight, theoretical pI and hydrophilia	ExPASy-Compute pl/Mw (http://web.expasy.org/compute_pi/ accessed on 15 October 2021)
Physical and chemical properties of protein	ExPASy-ProtParam (http://web.expasy.org/protparam/ accessed on 15 October 2021))
Hydrophilia	ExPASy-ProtScale (http://web.expasy.org/protscale/ accessed on 15 October 2021)
Functional site prediction of amino acid sequence	SoftBerryPSITE (http://www.softberry.com/berry.phtml?topic=psite&group=programs&subgroup=proloc/ accessed on 15 October 2021)
Protein transmembrane domain analysis	TMHMM Server v. 2.0 (http://www.cbs.dtu.dk/services/TMHMM accessed on 15 October 2021)
Predict potential signal peptide cleavage site	SignalP4.0-Server (http://www.cbs.dtu.dk/services/SignalP-4.0 accessed on 15 October 2021)
Analysis of protein structure and function domain	SMART (http://smart.embl-heidelberg.de/ accessed on 15 October 2021)
Prediction of protein secondary structure	Phyre^2^ (http://www.sbg.bio.ic.ac.uk/~phyre2/html/page.cgi?id=index accessed on 15 October 2021)
Automated 3D structure building	ExPASy-SWISS-MODEL (https://www.swissmodel.expasy.org/ accessed on 15 October 2021)
Multiple sequence alignment	ClustalW2 (http://www.ebi.ac.uk/Tools/msa/clustalw2/ accessed on 15 October 2021)
Reconstruct phylogenetic tree	MEGA6
Predict the subcellular localization	Plant-PLoc (http://www.csbio.sjtu.edu.cn/bioinf/Cell-PLoc/ accessed on 15 October 2021)

**Table 4 plants-12-01681-t004:** Three tubes were prepared for each reverse transcription product.

Amplification Program		Ingredient	Volume
95 °C 10 min		2xqPCRmix	10 μL
95 °C 10 s	40 cycles	F primer (10 pmol/uL)	0.5 μL
60 °C 60 s		R primer (10 pmol/uL)	0.5 μL
95 °C 5 s		DNA template	2 μL
60 °C 5 s	Test once every 0.5 °C temperature rise	ddH_2_O	7 μL
95 °C 5 s		total	20 μL

## Data Availability

The data presented in this study are available on request from the corresponding author.

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
