# Peer review of "Molecular Cloning and Expression Analysis of the Typical Class III Chitinase Genes from Three Mangrove Species under Heavy Metal Stress"

_plants, 2023, doi:10.3390/plants12081681_

Round 1

Reviewer 1 Report

The paper presents the results of cloning and analysis of the class III chitinase genes from three mangrove species and their defense role under heavy metal stress. The study of plant genes involved in the response to stress in plants is certainly a relevant and interesting topic for a wide range of readers. The work is mostly well thought out, implemented and written however there are a few remarks.

1.     Abstract. At the first mention of species, their full Latin names should be given.

2.     Abstract is not easy understandable. Line 17 – three different genes from three species and then in line 18 only one gene. Line 24 – what is C1, C2, C3, C4? In the abstract, it is better not to refer to the figures.

3.     Introduction. Lines 32-33 sentence looks unfinished.

4.     Line 54 - the abbreviation CBD must be deciphered

5.     Results. in Figure 1B, the gel looks upside down, there are no lane markings and no marker fragment lengths. BLAST cannot be performed with the obtained fragment, only with the nucleotide sequence of the obtained fragment.

6.     Figures 2B and 2C should be mentioned in the text.

7.     Lines 106-109 the sentence is inconsistent.

8.     Lines 116-118 the sentence sounds strange. (In this study there are metal binding sites and in other studies there are no such sites?)

9.     Figure 5 Morphological differences are not visible. Perhaps you need close-up photos of the leaves. It is not clear what C1, C2, C3, C4 are and what concentrations and what metals are in question. Perhaps it is worth adding a link to table 1, where metals and their concentrations are indicated.

10.  Figure 6. It is not clear what C1, C2, C3, C4 are and what concentrations and what metals are in question. Lack of deciphering the designations.

11.  Figure 7. Lack of deciphering the designations.

12.  Lines 158-159. Sentence needs to be revised.

13.  Discussion. Line 185 – Blast should be changed to BLAST.

14.  Line 190 – in vitro should be written in italics.

15.  Lines 192- 206. This paragraph refers to results, not discussion.

16.  The Discussion section lacks a discussion of Results part 2.2.

17.  Material and Methods. Line 276. E. coli should be written in italics.

Author Response

Response:

1) In the Abstract, three mangrove plants full Latin names have been given at the first mention of a species.

2) It has been cloned three different genes from three species. And the chitin proteins encoded by the three different genes are all proteins belonging to the same type, which is the typical class â…¢ chitinase with the characteristic catalytic structure belonging to the GH18 family and located outside the cell. There are some express errors in the article and they have been modified to “Bioinformatics analysis revealed that the three genes encoding proteins that all were all typical class â…¢ chitinases with the characteristic catalytic structure belonging to the GH18 family and located outside the cell”

C1, C2, C3, C4 representing the different treatment conditions. It has been deleted for K. obovate: C2, C4; R. stylosa: C1, C3, C4 in the Abstract.

3) Modified the sentence at lines 32-33. And we have revised the expressions to make them coherent and smooth.

4) Added the abbreviation CBD (Line 54).

5) Modified in Figure 1B, and have added lane markings and marker fragment lengths. BLAST cannot be performed with the obtained fragment, only with the nucleotide sequence of the obtained fragment, and revised the expressions to make them coherent and smooth.

6) Mentioned Figures 2B and 2C in the text “Through SMARTTM RACE cDNA amplification, the full-length cDNA sequence of chitinase was isolated from mangrove and named BgCHI III (figure 2A), KoCHI III (figure 2B), and RsCHI III (figure 2C).”

7) Modified the sentence in Lines 106-109.

8) Modified the sentence in Lines 116-118. In this study, there are metal binding sites and in other studies there are no such sites.

9) Thanking about the reviewer for pointing out this issue. Figure 5 Morphological differences are not visible of B. gymnorrhiza, and the leaves of K.obovate, R. stylosa  lose too much water and become wilted. But sorry that we don’t have close-up photos of the leaves.

The mangrove seedlings were irrigated with 1/2 Hoagland’s nutrient solution containing mixed heavy metal sewage of different pollution levels (Table 1). C1: 5 mg/L CuCl2, 1 mg/L PbCl2, 0.2 mg/L CdCl2; C2: 25 mg/L CuCl2, 5 mg/L PbCl2, 1 mg/L CdCl2; C3: 50 mg/L CuCl2, 10 mg/L PbCl2, 2 mg/L CdCl2; C4: 75 mg/L CuCl2, 15 mg/L PbCl2, 3 mg/L CdCl2. We added the detailed information in the Materials and Methods. The selection of heavy metals and the setting of concentrations were determined by checking the relevant literature.

10) C1: 5 mg/L CuCl2, 1 mg/L PbCl2, 0.2 mg/L CdCl2.

11) Added related description the results of Figure 7 in section 2.3.

12) Modified the sentence in Lines 158-159.

13) Changed Blast to BLAST.

14) Modified in vitro to be written in italics.

15) It has been put the content of this paragraph in the results section 2.1 and the Figure 8 have changed to Figure 5. The numbering of the pictures after Figure 5 is changed.

16) Added more to the Discussion section’s discussion of the Results part 2.2.

17) Modified the E. coli should be written in italics.

Reviewer 2 Report

The article, "Molecular cloning and expression analysis of the typical class â…¢ chitinase genes from three mangrove species under heavy metal stress" is well written but some changes are required,

1) English needs to be improved; there are many grammatical and spelling mistakes. I recommend taking an English editing service.

2) Abstract and introduction are too short. Some more details need to be added to the abstract. The introduction should be more elaborative with a review; the study's objective is missing. 

3) Discussion is fragmented, with no link between the studied attributes.

4) The statistical analysis needs more clarification. 

5) Figure 6 and 7 need to be reconstructed again. Lettering should be done on significant results and p values must be added with statistically significant values. Figure 7C missing legends in the figure. Authors must add all bars in the figures no matter if the values were zero. 

Author Response

Response:

1) English has been improved, and involved native English speakers for language corrections.

2) It has been enriched for the Abstract to reach the manuscript requirement of shorter than 200 words. It has also been added for the introduction to be more elaborative with a review; the study's objective is missing. the manuscript content to exceed more than 4000 words.

3) Thanking about the reviewer for pointing out this issue, and added link between the studied attributes in discussion and added the Figure 9.

4) The statistical analysis has added more clarification in 4.7 Statistics in Materials and Methods.

5) Figure 6 and 7 need to be reconstructed again.

Round 2

Reviewer 2 Report

The authors made all the changes in the manuscript, and the article is fit for possible publication in this journal.